# Deep Non-Blind Deconvolution via Generalized Low-Rank Approximation

**Wenqi Ren** *
IIE, CAS

**Jiawei Zhang**
SenseTime Research

**Lin Ma**
Tencent AI Lab

**Jinshan Pan**
NJUST

**Xiaochun Cao** †
IIE, CAS

**Wangmeng Zuo**
HIT

**Wei Liu**
Tencent AI Lab

**Ming-Hsuan Yang**
UCMerced, Google Cloud

## Abstract

In this paper, we present a deep convolutional neural network to capture the inherent properties of image degradation, which can handle different kernels and saturated pixels in a unified framework. The proposed neural network is motivated by the low-rank property of pseudo-inverse kernels. Specifically, we first compute a generalized low-rank approximation to a large number of blur kernels, and then use separable filters to initialize the convolutional parameters in the network. Our analysis shows that the estimated decomposed matrices contain the most essential information of an input kernel, which ensures the proposed network to handle various blurs in a unified framework and generate high-quality deblurring results. Experimental results on benchmark datasets with noisy and saturated pixels demonstrate that the proposed deconvolution approach relying on generalized low-rank approximation performs favorably against the state-of-the-arts.

## 1 Introduction

Image blur is often inevitable due to numerous factors including low illumination, camera motion, telephoto lens, or small aperture for a wide depth of field. The shift-invariant blur process can be modeled by

$$y = c(k * x) + n, \tag{1}$$

where $y$, $x$, $k$, and $n$ denote blurry input, latent image, blur kernel, and image noise, respectively; $*$ denotes a convolution operator; $c(\cdot)$ is a non-linear function describing a camera imaging system. It is well-known that estimating the latent image from a blurry input is challenging. If the blur kernel is unknown, the problem is called blind deconvolution [8, 19]. Otherwise, it reduces to non-blind deconvolution [9, 26] if the blur kernel is known. As non-blind deconvolution remains as an active and challenging research topic due to its ill-posedness [32], we present a method to tackle this problem.

Existing algorithms are usually based on the spatial [5, 6, 21] or frequency [3, 15, 16] domain. However, the spatial domain based methods have a high computational cost since these methods need to solve large linear systems. Although the frequency-based approaches are computationally efficient thanks to the use of Fast Fourier Transformations (FFTs), these methods often generate significant ringing artifacts since blur kernels are band-limited with sharp frequency cut-off. In addition, existing non-blind deconvolution algorithms usually assume that the noise level is small and less effective for blurry images with significant noisy and saturated pixels [31].

To solve the aforementioned issues, Xu *et al.* [31] propose a deep convolutional neural network (CNN) by combining spatial deconvolution and CNNs to overcome the drawbacks of existing deconvolution methods. However, Xu *et al.*'s method needs to retrain the network for different blur kernels, which is not practical in real-world scenarios. For instance, it is necessary to train multiple models for cameras with different lenses or apertures.

Inspired by the low-rank property of pseudo-inverse kernels, we propose a generalized deep CNN to handle arbitrary blur kernels in a unified framework without re-training for each kernel as in [28, 31]. Different from previous learning based approaches [13, 37], our approach does not require any pre-processing to deblur images. Instead, we initiate the image deconvolution process based on a low-rank approximation to a large number of blur kernels. In contrast to existing CNN-based methods [37, 39] that directly learn mappings from blurred inputs to sharp outputs, we propose a novel strategy to properly initialize the weights in the network capitalizing on Generalized Low Rank Approximations (GLRAs) of kernel matrices, which cannot be easily achieved by the conventional training procedures based on random initialization. Experimental results show that our approach performs favorably against other state-of-the-art non-blind deconvolution methods, especially when the blurred images contain significant noisy and saturated pixels.

The contributions of this work are summarized as follows.

- We establish the connection between optimization schemes and CNNs, and propose an image deconvolution approach by using the separable structure of kernels to initialize the weights in the network, which can be generalized to arbitrary blur kernels.

- We analyze the low-rank property of various kernel types and sizes, which is the key point of the unified deconvolution network that can model arbitrary kernels.

- We quantitatively evaluate the proposed approach against the state-of-the-art methods. The results along with analysis show that significant ringing artifacts and visual artifacts can be effectively reduced by the proposed approach especially when blurred images retain noisy and saturated pixels.

## 2   Related Work

Non-blind deconvolution has attracted much attention with significant advances [10, 11, 24] in recent years due to its importance in computer vision and machine learning. Existing methods can be roughly categorized into spatial domain based methods using statistical image priors, frequency-based methods, and data-driven schemes.

**Deconvolution in the spatial domain based on statistical image priors.** As non-blind deconvolution is an ill-posed problem, most existing methods make assumptions on the latent images based on statistical priors [2, 33, 36]. To suppress ringing artifacts, sparse image priors have been proposed to constrain the solution space, *e.g.*, hyper-Laplacian image priors [12, 15, 17]. Schmidt *et al.* [27] use a Bayesian minimum mean squared error estimate and the fields of experts framework [23] to model image priors. Cho *et al.* [4] develop a variational EM approach to remove saturation regions with a Gaussian likelihood function.

The Gaussian mixture model (GMM) has also been used to fit the distribution of natural image gradients. In [6] Fergus *et al.* use a GMM to learn an image gradient prior via variational Bayesian inference. Zoran and Weiss [41] propose a patch-based prior following a GMM, which is further extended with a multi-scale patch-pyramid model [29]. On the other hand, Roth and Black propose a non-blind deconvolution method based on a field of experts [23]. However, all these spatial domain based deconvolution methods are computationally expensive.

**Deconvolution in the frequency domain using FFTs.** Early frequency-based method, *e.g.*, Richardson-Lucy method [22] and Weiner filtering [30], tend to generate considerable artifacts in the recovered images. Due to the computational efficiency, non-blind deconvolution algorithms in the frequency domain using the half-quadratic splitting scheme are proposed [15] in the literature.

However, frequency domain based deconvolution methods are less effective in handling irregular regions due to the band-limited property caused by cutting off in the frequency domain. At these frequencies, the direct inverse of a kernel usually has a large magnitude and amplifies signal and noise significantly. After the deconvolution process, it is difficult to remove these artifacts.

**Data-driven deconvolution schemes.** Numerous image restoration algorithms counting on CNNs have recently been proposed [20, 35, 38, 40]. In [28], deep networks are used to learn the mapping functions from corrupted patches to clean patches. Xu *et al.* [31] establish the connection between optimization-based schemes and neural networks, and develop an efficient method based on singular value decomposition (SVD) to initialize the network weights. However, these methods need to re-train the network for different kernels, which cannot be applied to real-world scenarios. While some efforts have been made in handling multiple kernels in a single network [37, 39], the priors related to blur kernels have not yet been used to constrain the mapping space.

Different from those aforementioned methods, we address the problem of non-blind deconvolution by exploiting a generalized low-rank approximation of blur kernels, and improve the deblurring performance across convolutional layers.

## 3 Proposed Algorithm

In this section, we first illustrate the separability of blur kernels, and then propose a neural network capitalizing on the low-rank property of pseudo-inverse kernels.

### 3.1 Separability for A Single Kernel

To better understand the separability of blur kernels, we first consider the following simple linear convolution model $y = k * x$. Based on the Fourier theory, the spatial convolution can be transformed to the frequency-domain multiplication by

$$\mathcal{F}(y) = \mathcal{F}(k) \circ \mathcal{F}(x), \tag{2}$$

where $\mathcal{F}(\cdot)$ denotes the discrete Fourier transform and $\circ$ is an element-wise multiplication. In the frequency domain, $x$ can be obtained as

$$x = \mathcal{F}^{-1}(1/\mathcal{F}(k)) * y = k^\dagger * y, \tag{3}$$

where $k^\dagger$ is the spatial pseudo-inverse kernel. The singular value decomposition (SVD) of $k^\dagger$ can be obtained by

$$k^\dagger = USV^\top = \sum_j s_j \cdot u_j * v_j^\top, \tag{4}$$

where $u_j$ and $v_j$ denote the $j$-th columns of U and V, respectively, and $s_j$ is the $j$-th singular value. We note that using the decomposed $u_j$ and $v_j$ as the weight initialization in CNNs would lead to a more expressive network for image deconvolution [31]. However, the model in (4) is only applicable to a single kernel, and the network needs to be retrained when the blur kernel changes. Consequently, this increases the complexity and difficulty for practical applications as blur kernels are of a great variety. In the following, we propose an approach relying on low-rank approximation of matrices to tackle this problem.

### 3.2 Separability for A Large Number of Kernels

To avoid retraining the network for each blur kernel, we propose a separability approach for a large number of kernels and construct a unified network to learn the high-dimensional mapping.

Let $\{k_p^\dagger\}_{p=1}^n \in \mathbb{R}^{d \times d}$ be a set of pseudo-inverse kernels, where $d$ denotes the size for each inverse kernel and $n$ is the number of pseudo-inverse kernels. We aim to compute matrices $L \in \mathbb{R}^{d \times m}$, $R \in \mathbb{R}^{d \times m}$ and matrices $\{M_p\}_{p=1}^n \in \mathbb{R}^{m \times m}$, so that $LM_pR^\top$ can approximate an arbitrary pseudo-inverse kernel $k_p^\dagger$, where the columns in $L \in \mathbb{R}^{d \times m}$ and $R \in \mathbb{R}^{d \times m}$ are orthogonal, and $d$ and $m$ are pre-specified parameters based on empirical results.

To obtain the matrices $L$, $R$ and $\{M_p\}_{p=1}^n$, we solve a minimization problem as

$$\min_{L,R,M_p} \sum_{p=1}^n \|k_p^\dagger - LM_pR^\top\|_F^2. \tag{5}$$

The matrices $L$ and $R$ in (5) operate as the two-sided linear transformations on a large set of kernels. With the estimated matrices $L$, $R$, and $\{M\}_{p=1}^n$, we can recover the original pseudo-inverse kernel $k_p^\dagger$ by $LM_pR^\top$ for each $p$. In this paper, we employ the generalized low-rank approximations

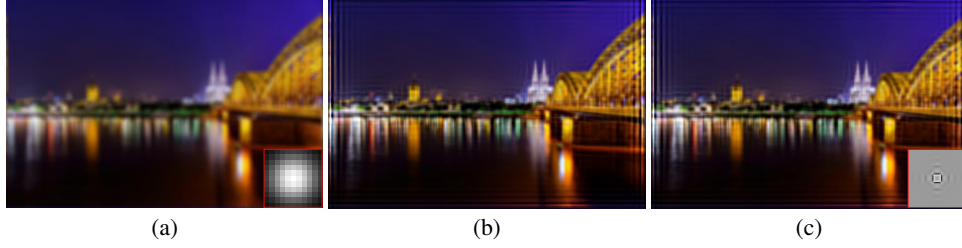

(a)                      (b)                      (c)

Figure 1: A deconvolution example of the pseudo-inverse kernel by GLRA. (a) A blurred image and a Gaussian kernel. (b) Deblurred result by the inverse kernel with the size of $300 \times 300$. (c) Deblurred result by the estimated inverse kernel using GLRA in (8).

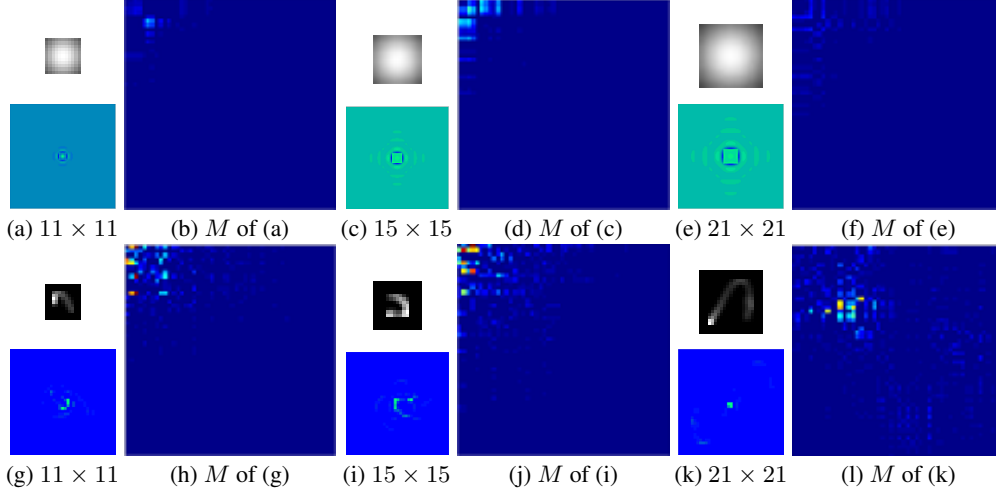

(a) $11 \times 11$    (b) $M$ of (a)    (c) $15 \times 15$    (d) $M$ of (c)    (e) $21 \times 21$    (f) $M$ of (e)

(g) $11 \times 11$    (h) $M$ of (g)    (i) $15 \times 15$    (j) $M$ of (i)    (k) $21 \times 21$    (l) $M$ of (k)

Figure 2: Matrices $M$ (with size $50 \times 50$) of different kernel types and sizes. Top row: Gaussian kernels, pseudo-inverse kernels, and matrices $M$. Bottom row: Motion kernels, pseudo-inverse kernels, and matrices $M$. The non-zero number increases as the kernel size is larger. The values of $M$ of Gaussian kernels are mainly distributed on the upper-left borders, while the values of $M$ of motion kernels are mainly distributed on the diagonal.

(GLRA) method [34] to compute the matrices $L$ and $R$. In contrast to SVD that converts a single matrix $k^\dagger$ to vectors, GLRA directly manipulates pseudo-inverse kernels $\{k^\dagger\}_{p=1}^n$ and computes two transformations $L = [l_1, l_2, \ldots, l_m]$ and $R = [r_1, r_2, \ldots, r_m]$ with orthogonal columns.

Given a set of spatial pseudo-inverse kernels $\{k^\dagger\}_{p=1}^n$, we can decompose these kernels by

$$k_p^\dagger = L M_p R^\top = \sum_{i,j} l_i * M_{p_{i,j}} * r_j^\top, \tag{6}$$

where $M_{p_{i,j}}$ denotes the pixel at the $i$-th row and $j$-th column of $M_p$. Therefore, given a testing pseudo-inverse kernel $k_t^\dagger$, we can first compute $M$ by

$$M = L^\top k_t^\dagger R. \tag{7}$$

Then we can estimate the sharp image $x$ by convolving $k_t^\dagger$ with the blurred image $y$:

$$x = k_t^\dagger * y = \sum_{i,j} l_i * M_{i,j} * r_j^\top * y, \tag{8}$$

which shows that 2D deconvolution can be regarded as a weighted sum of separable 1D filters ($l_i$ and $r_j$). In practice, we can approximate $k_t^\dagger$ well by a small number of separable filters by dropping out the kernels associated with zero or small $M_{i,j}$.

Figure 1(a) shows a blurred image convolved by a Gaussian kernel. In Figure 1(b), we first show the deblurred result by the inverse kernel with a large size of $300 \times 300$. The estimated pseudo-inverse kernel and the deblurred result are shown in Figure 1(c). The deblurred result by the separable filter in (8) is close to that in Figure 1(b), which demonstrates the effectiveness of the GLRA method.

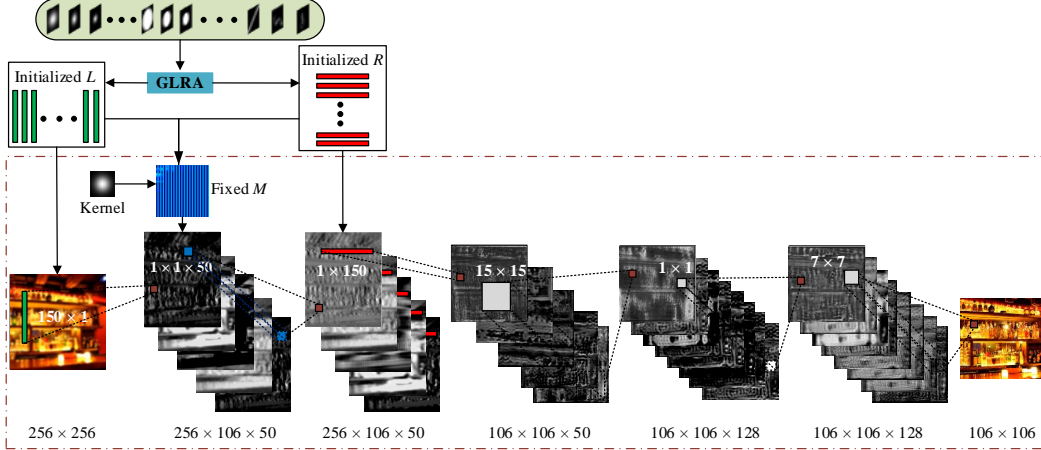

Figure 3: The architecture of the proposed deconvolution network. We use the separable filters ($l_i$ and $r_i$) of a large number of blur kernels by GLRA to initialize the parameters of the first and third layers, and use the estimated $M$ for each blur kernel to fix the parameters in the second convolutional kernels. The three more convolutional layers are stacked in order to remove artifacts.

**Property of $M$ for Different Kernels.** Note that the matrix $M_p$ in (6) is not required to be diagonal. We find that the distribution of the elements in $M$ depends on certain kernel types and sizes. As shown in Figure 2(a)-(f), elements of $M$ with large values mainly distribute on the upper-left borders if the type of the blur kernel is Gaussian. In contrast, elements of $M$ with large values mainly distribute on the diagonal if the input is a motion kernel as shown in Figure 2(g)-(l). In addition, the number of elements in $M$ with large values increases as the size of the blur kernel increases. Therefore, the matrix $M$ contains the most essential information of the input blur kernel. This is the main reason that the proposed approach can handle arbitrary kernels in a unified network.

### 3.3 Network Architecture

We design the convolutional network based on the kernel separability theorem in Section 3.2. The proposed network architecture is shown in Figure 3. The first three convolutional layers are the deconvolution block. We use the separable filters ($l_i$ and $r_j$) generated by GLRA in (6) to initialize the weights in the first and third convolutional kernels. The feature maps in the first and third layers are thus generated by applying $m$ one-dimensional kernels of sizes $d \times 1$ and $1 \times d$, respectively. For each pair of blurred image and kernel, we use (7) to compute the corresponding $M$ and set the $m$ columns $M_j$ as the parameters of $m$ kernels of size $1 \times 1 \times m$ in the second layer. Empirically, we find that an inverse kernel with size of 150 is typically sufficient to generate visually plausible deconvolution results, and that a matrix $M$ with size of $50 \times 50$ contains the most values larger than zero. Thus, we set $m = 50$ and $d = 150$ in this paper. More analysis about these two parameters can be found in Section 5.2.

For image deconvolution, there are several merits for using the initialization by GLRA. First, the generalized low-rank property enables the network to handle arbitrary kernels in a unified network. Second, the separability of kernels for deconvolution can effectively constrain the mapping space. Third, the low-rank property of pseudo-inverse kernels makes the network more expressive and compact than conventional CNN-based networks [35, 37]. In addition, to handle saturations, we add three more convolutional layers to remove ringing artifacts as in [31]. We set the sizes of these three convolutional filters to $15 \times 15$, $1 \times 1$, and $7 \times 7$, respectively. While the number of weights grows due to the additional layers, it facilitates handling complex outliers and artifacts in image deblurring.

## 4 Experimental Results

We evaluate the proposed approach against the state-of-the-art non-blind deconvolution methods including hyper-Laplacian (HL) prior [15], expected patch log-likelihood (EPLL) [41], variational EM (VEM) [4], multi-layer perceptron (MLP) [28], cascade of shrinkage fields (CSF) [25], deep convolutional neural network (DCNN) [31], deep CNN denoiser prior (IRCNN), and fully convolutional networks (FCNN) [37]. For fair comparisons, we use the original implementations of these methods

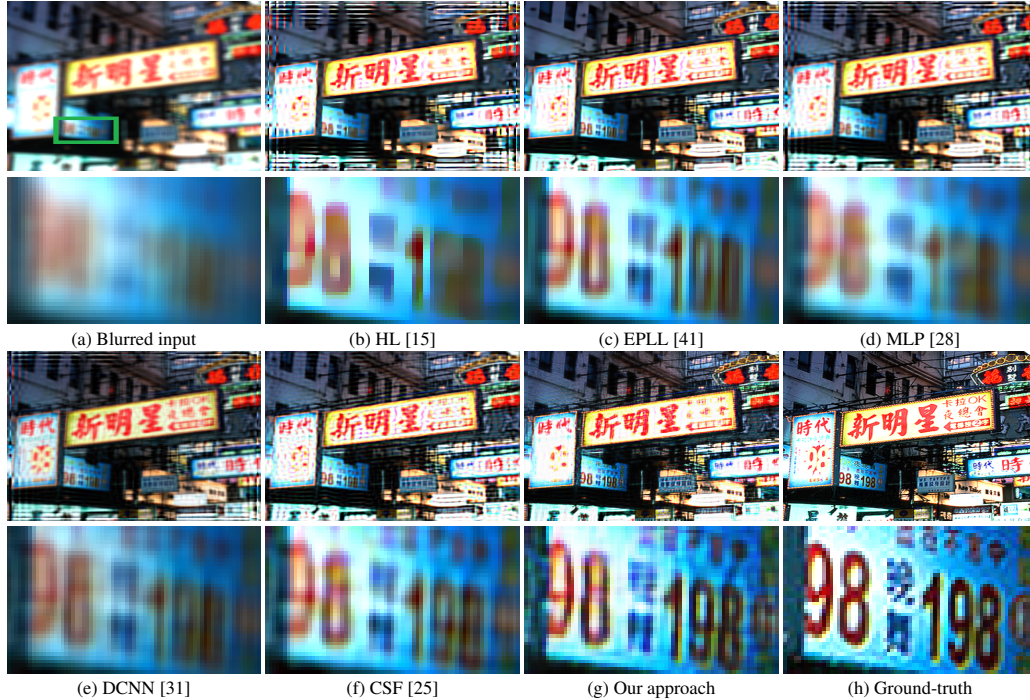

|  |  |  |  |
|:---:|:---:|:---:|:---:|
| (a) Blurred input | (b) HL [15] | (c) EPLL [41] | (d) MLP [28] |
| (e) DCNN [31] | (f) CSF [25] | (g) Our approach | (h) Ground-truth |

Figure 4: Visual comparisons of deconvolution results of Gaussian blur. The results by HL [15], MLP [28], and DCNN [31] methods tend to generate ringing artifacts. The deblurred results generated by EPLL [41] and CSF [25] schemes still contain some blurs. In contrast, the deblurred image obtained by the proposed approach is closer to the ground-truth.

Table 1: Average PSNR and SSIM on the evaluation image set.

|  | Random | HL [15] | EPLL [41] | VEM [4] | MLP [28] | CSF [25] | DCNN [31] | FCNN [37] | Our approach |
|---|---|---|---|---|---|---|---|---|---|
| | | | | Gaussian blur with saturated pixels | | | | | |
| PSNR | 22.8520 | 23.2764 | 24.2021 | 24.0954 | 21.8684 | 23.9879 | 23.8653 | 23.6058 | **25.6931** |
| SSIM | 0.7016 | 0.7675 | 0.8754 | **0.8822** | 0.7948 | 0.8543 | 0.7098 | 0.7384 | 0.8768 |
| | | | | Disk blur with saturated pixels | | | | | |
| PSNR | 21.1734 | 23.0128 | 24.0970 | 23.7499 | 22.3761 | 22.9271 | 22.8102 | 21.8805 | **24.4988** |
| SSIM | 0.7529 | 0.8563 | 0.8754 | 0.8793 | 0.8385 | 0.8319 | 0.7508 | 0.8336 | **0.8851** |

and tune the parameters to generate the best possible results. The implementation code, the trained model, as well as the test data, can be found at our project website.

## 4.1 Network Training

The image patch size is set as $256 \times 256$ in the proposed network. We use the ADAM [14] optimizer with a batch size 1 for training with the $L_2$ loss. The initial learning rate is 0.0001 and is decreased by 0.5 for every 5,000 iterations. Note that we fix parameters in the second layer from the estimated $M$ without tuning the parameters. The first three layers are trained using the initialization from separable inversion as described Section 3.3. We use the Xavier initialization method [7] to set the weights of the last three convolutional kernels. For all the results reported in the paper, we train the network for 200,000 iterations, which takes 30 hours on an Nvidia K80 GPU. The default values of $\beta 1$ and $\beta 2$ (0.9 and 0.999) are used, and we set the weight decay to 0.00001.

## 4.2 Dataset

**Training data.** In order to generate blurred images for training, we use the BSD500 dataset [1] and randomly crop image patches with a size of $256 \times 256$ pixels as clear images. We use Gaussian,

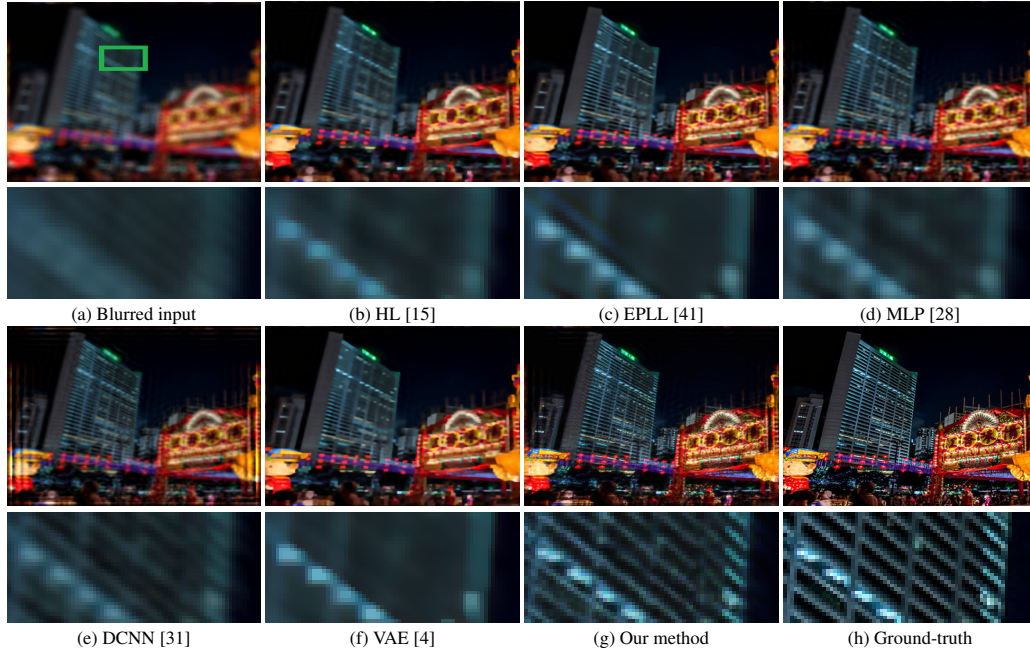

|  | (a) Blurred input | (b) HL [15] | (c) EPLL [41] | (d) MLP [28] |
|  | (e) DCNN [31] | (f) VAE [4] | (g) Our method | (h) Ground-truth |

Figure 5: Visual comparisons of deconvolution results of a disk blur. The deblurred results in (b)-(f) contain ringing artifacts and residual blurs (best viewed on a high-resolution display).

Table 2: Average PSNR and SSIM on the BSD100 testing dataset [18].

|  | HL [15] | EPLL [41] | VEM [4] | MLP [28] | CSF [25] | IRCNN[39] | FCNN [37] | Our method |
|---|---|---|---|---|---|---|---|---|
| *Gaussian blur with saturated pixels* | | | | | | | | |
| PSNR | 21.88 | 21.9068 | 21.8034 | 21.8164 | 21.4394 | 22.3735 | 21.6209 | **23.2141** |
| SSIM | 0.6194 | 0.7756 | 0.7806 | 0.7701 | 0.7641 | **0.8012** | 0.7673 | 0.7730 |
| *Disk blur with saturated pixels* | | | | | | | | |
| PSNR | 21.5779 | 22.7244 | 22.6630 | 22.2198 | 22.0775 | 24.0907 | 21.7993 | **24.2379** |
| SSIM | 0.6101 | 0.8181 | 0.8235 | 0.7955 | 0.7856 | **0.8783** | 0.7822 | 0.8147 |

disk, and motion kernels for performance evaluation. The motion kernels are generated according to [37], and the blur kernel size ranges from 9 to 27 pixels. We convolve clear image patches with blur kernels and add $1\%$ Gaussian noise to generate blurred image patches. To synthesize saturated regions, we first enlarge range of both blurred and clear images by a factor of 1.2, and then clip the images into the dynamic range of 0 to 1.

**Testing data.** For the test dataset, we first download 30 ground-truth clear images from Flickr, and then generate 30 different Gaussian kernels and 30 disk kernels to synthesize blurry images. Then, we evaluate the proposed algorithm on the BSD100 testing dataset [18] blurred by 100 random Gaussian kernels and 100 disk kernels. We also add $1\%$ noise and saturated pixels in the blurred images to evaluate the performance of the deconvolution methods.

### 4.3 Defocus Blur

Similar to the state-of-the-art algorithms, we quantitatively evaluate the proposed method on the blurred images degraded by Gaussian and disk blurs, which are commonly used to model defocus blur.

**Gaussian blur.** We first evaluate the proposed method on the dataset degraded by Gaussian kernels with $1\%$ noise. As shown in Table 1, the propsoed method performs well against the HL [15], EPLL [41], MLP [28], CSF [25], DCNN [31] and FCNN [37] schemes in terms of PSNR and SSIM. Although VAE [4] method performs slightly better than the proposed methods in terms of SSIM, our method achieves performance gain of 1.6 dB in terms of PSNR when compared with the VAE [4]

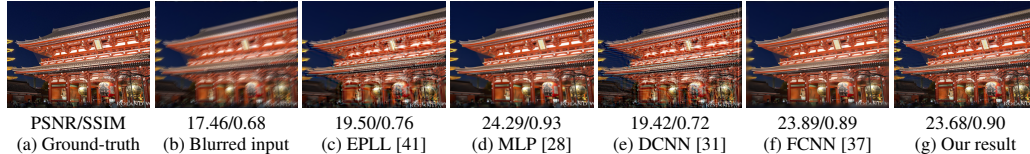

| | | | | | | |
|---|---|---|---|---|---|---|
| PSNR/SSIM | 17.46/0.68 | 19.50/0.76 | 24.29/0.93 | 19.42/0.72 | 23.89/0.89 | 23.68/0.90 |
| (a) Ground-truth | (b) Blurred input | (c) EPLL [41] | (d) MLP [28] | (e) DCNN [31] | (f) FCNN [37] | (g) Our result |

Figure 6: Visual comparisons of deconvolution results of motion blur. The proposed method performs favorably compared with existing non-blind deconvolution methods.

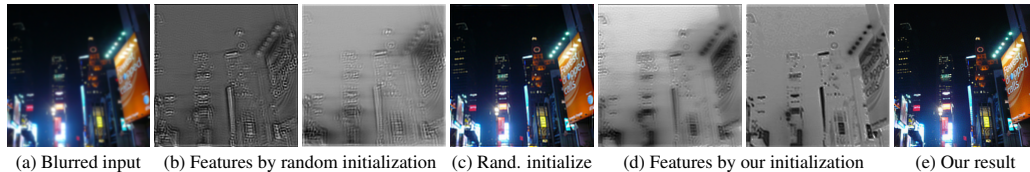

| (a) Blurred input | (b) Features by random initialization | (c) Rand. initialize | (d) Features by our initialization | (e) Our result |
|---|---|---|---|---|

Figure 7: Comparisons of feature maps from the 4-th and 5-th layers. (b) Feature maps from random initialization. (d) More informative maps using our initialization scheme. (c) and (d) are the results by random initialization and our approach (best viewed on high-resolution displays).

method. In addition, we show deblurred images by the evaluated methods in Figure 4. The results by the HL [15], MLP [28] and DCNN [31] methods contain some ringing artifacts. On the other hand, the EPLL [41] and CSF [28] algorithms fail to generate clear images. In contrast, the deblured image by the proposed method has clearer textures (See Figure 4(g)). Table 2 also demonstrates that the proposed algorithm performs favorably against the state-of-the-art methods on the BSD100 testing dataset. We note that although the SSIM value by IRCNN [39] is 0.03 higher than our method, our method achieves better results by up to 0.84 dB in terms of PSNR.

**Disk blur.** We further evaluate our method on the blurred images degraded by disk kernels and 1% noise. Table 1 shows that the proposed algorithm achieves better performance compared to the state-of-the-art methods. Figure 5(g) demonstrates that our algorithm generates more visually pleasant results than other deconvolution methods. The results in Table 2 also show that our algorithm performs favorably against the non-blind deconvolution approaches on the BSD100 dataset.

## 4.4 Motion Blur

In this section, we show that the proposed method is good at non-blind deconvolution for images degraded by motion blurs. As analyzed in Section 3.2, the matrix $M$ has different properties for different kernel types and sizes, which makes it feasible to handle arbitrary kernels in a unified network. As shown in Figure 6, the generated result by the EPLL [41] method still contains blurry artifacts since this method cannot handle blurred images with saturated pixels. Compared to the state-of-the-art CNN-based methods [31, 37], the deblurred image by our proposed algorithm is sharper, which demonstrates that the use of GLRA in neural networks is effective for image deconvolution. We note that MLP [28] generates the result with higher PSNR and SSIM values. The main reason is that the rank of motion kernels is higher than that for the Gaussian and disk kernels. Our future work will address this issue with more motion kernel priors.

## 5 Analysis and Discussions

In this section, we analyze how the GLRA based initialization method helps estimate clear scenes and present sensitivity analysis with respect to the parameter settings and noises.

## 5.1 Effectiveness of The Proposed Initialization Method

As the optimization function for a deep CNN is highly non-convex, training the whole network with random initialization is less effective and usually converges to a poor local minimum. As a result, the trained model with random initial weights is not effective in removing image blurs discussed in this work. To better understand the importance of initialization, we analyze the feature maps from the last two layers in the proposed CNN. Some sample results are shown in Figure 7, where (a) is a blurred input, (b) is the feature maps from the 4-th and 5-th layers by random initialization, respectively, and

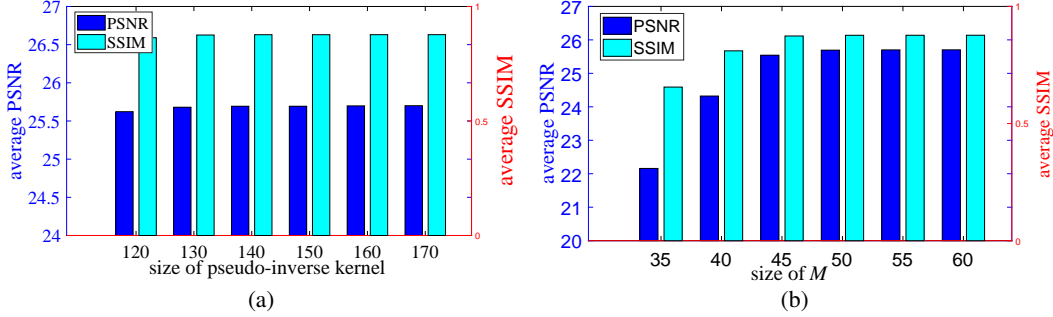

Figure 8: Sensitivity analysis with respect to parameters $d$ and $m$.

(c) is the deblurred result by random initialization. The maps in (b) contain blurry boundaries, which indicates that an algorithm with random initialization is unlikely to deblur images effectively. In contrast, the maps in (d) show clear edges and result in a sharper and visually more pleasant deblurred image in (e).

## 5.2 Parameter Analysis

The proposed deconvolution model involves two main parameters, *i.e.*, size $d$ of pseudo-inverse kernel and size $m$ of matrices $M$. In this section, we evaluate the effect of these parameters on image deblurring using the testing dataset. For each parameter, we carry out experiments with different settings by varying one and fixing the others, and use PSNR and SSIM to measure the accuracy. Figure 8 shows that the proposed deconvolution algorithm is insensitive to parameter settings.

## 5.3 Sensitivity to the Noise

In addition to the testing data with 1% Gaussian noise in Section 4, we further evaluate our method on the images with 2% and 3% Gaussian noises. Table 3 shows that the proposed method performs well even when the noise level is high, which demonstrates that the proposed algorithm is more robust to noise than the state-of-the-art methods.

Table 3: Average PSNR and SSIM for 2% and 3% noises.

| HL [15] | MLP [28] | CSF [25] | FCNN [37] | Ours | EPLL [41] | DCNN [31] | CSF [25] | FCNN [37] | Ours |
|---|---|---|---|---|---|---|---|---|---|
| | | 2% noise | | | | | 3% noise | | |
| 20.72/0.61 | 20.64/0.70 | 20.13/0.68 | 20.45/0.70 | **22.15/0.70** | 22.60/0.74 | 22.54/0.71 | 21.95/0.71 | 22.42/0.74 | **23.53/0.74** |

## 6 Concluding Remarks

In this work, we propose a deconvolution approach relying on generalized low-rank approximations of matrices. Our network exploits the low-rank property of blur kernels and deep models by incorporating generalized low-rank approximations of pseudo-inverse kernels into the proposed network model. We analyze the property of the decomposed variable $M$ in GLRA for different kernels to demonstrate that the proposed approach can handle arbitrary kernels in a unified framework. In addition, our analysis shows that the deep CNN initialized by GLRA is able to avoid poor local minimum and benefit blur removal. The experimental results demonstrate that the proposed approach achieves favorable performance against the state-of-the-art deconvolution methods.

## Acknowledgment

This work is supported in part by the National Key R&D Program of China (Grant No. 2016YFC0801004), National Natural Science Foundation of China (No. 61802403, U1605252, U1736219, 61650202), Beijing Natural Science Foundation (No.4172068). W. Ren is supported in part by the Open Projects Program of National Laboratory of Pattern Recognition and the CCF-Tencent Open Fund. J. Pan is supported in part by the Natural Science Foundation of Jiangsu Province (No. BK20180471). M.-H. Yang is supported in part by the NSF CAREER Grant #1149783 and gifts from and NVIDIA.

## Footnotes

*Part of this work was done while Wenqi Ren was with Tencent AI Lab as a Visiting Scholar.

†Corresponding author, caoxiaochun@iie.ac.cn.

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
