[Reviews · NeurIPS 2018]

Reviewer 1



This paper extends the non-blind deconvolution in [1]. The author proposes to use a generalized low-rank approximation method to model a wide range of blur kernels and train the non-blind deconvolution network with the low-rank representation. Strength: - Unlike the model in [1] that requires a specific training (or fine tuning) for each kernel, the proposed model can handle different blur kernels. - The model can use the low-rank representation results to initialize the model parameters. - Experiments show that the model can achieve good performances on different images and performs better than the compared methods, such as [1]. Weakness: - The most important motivation of this paper is to make the learning-based CNN model being able to handle different kernels. I did not find the reason for using the low-rank approximation-based method. Regarding the extension of the model in [1], I agree the low-rank representation idea is straightforward and clever, but, generally, I am wondering whether building a pre-defined and restricted representation for kernels is necessary. In many optimization-based neural network design, the blur kernels can be directly fed into a learnable neural network. Please check [2,3]. They both can work as a flexible model for different kernels and perform very well for different images. I want to see the discussions and comparisons (if possible) with these neural networks motivated by the conventional optimizer. - The author may experiment to study the sensitivity of the proposed model to the noise. - More experiments on the standard non-blind deconvolution benchmark datasets are necessary to validate the proposed method. [1] L. Xu, J. S. Ren, C. Liu, and J. Jia. Deep convolutional neural network for image deconvolution. In NIPS, 2014. [2] Kruse, Jakob, Carsten Rother, and Uwe Schmidt. "Learning to push the limits of efficient fft-based image deconvolution." IEEE International Conference on Computer Vision (ICCV). 2017. [3] Gong, D., Zhang, Z., Shi, Q., Hengel, A. V. D., Shen, C., & Zhang, Y. (2018). Learning an Optimizer for Image Deconvolution. arXiv preprint arXiv:1804.03368. [4] Zhang, K., Zuo, W., Gu, S., & Zhang, L. (2017, July). Learning deep CNN denoiser prior for image restoration. In IEEE Conference on Computer Vision and Pattern Recognition (Vol. 2). The authors address my major concerns. I tend to vote for acceptance.

Reviewer 2



In this paper, the author present a deep convolutional neural network to capture the inherent properties of image degradation. The proposed neural network is motivated by the low-rank property of pseudo-inverse kernels. The analysis shows that the estimated decomposed matrices contain the most essential information of the input kernel. Experimental results on benchmark datasets with noise and saturated pixels demonstrate that the proposed algorithm performs favorably against state-of-the-art methods. The paper is well-organized and written. The proposed model can handle different types of blur kernels. Also, the network is well validated by Discussion Section and insensitive to the parameters. In my opinion, the author introduce the off-the-shelf GLRA method to approximate a set of blur kernels to obtain the initializations of the network parameters. It's more like a improved variant of the previous work by Xu et.al. [22]. The novelty is somehow weak.

Reviewer 3



In this paper the authors propose the use of generalized low rank approximations for approximating the psuedo inverse kernels for non-blind deconvolution. The authors compare their methods with a number of other methods. The method is illustrated on a few sample images and shows quantitatively and qualitatively better results. The proposed work provides evaluation only on a few (10-12) sample images in the main paper and supplementary. One way for evaluating it on a large scale is by considering a standard benchmark dataset such as BSD or some such and yevaluating the method on a few hundred images and providing statistically significant improvement results. This would make the method to be more acceptable and to do this with a number of varying kernel sizes The current method is suited for non-blind deconvolution. One question therefore is whether it would be tolerant to errors in estimation of the kernel. It would be great if some analysis to that extent could be provided.